# Novel Insights and Mechanisms of Lipotoxicity-Driven Insulin Resistance

**DOI:** 10.3390/ijms21176358

**Published:** 2020-09-02

**Authors:** Benjamin Lair, Claire Laurens, Bram Van Den Bosch, Cedric Moro

**Affiliations:** 1INSERM, UMR1048, Institute of Metabolic and Cardiovascular Diseases, 31432 Toulouse, France; benjamin.lair@inserm.fr (B.L.); claire.laurens@inserm.fr (C.L.); bram.vandenbosch@student.kuleuven.be (B.V.D.B.); 2University of Toulouse, Paul Sabatier University, 31330 Toulouse, France

**Keywords:** type 2 diabetes, skeletal muscle, liver, adipose tissue, diacylglycerols, ceramides, insulin signaling

## Abstract

A large number of studies reported an association between elevated circulating and tissue lipid content and metabolic disorders in obesity, type 2 diabetes (T2D) and aging. This state of uncontrolled tissue lipid accumulation has been called lipotoxicity. It was later shown that excess lipid flux is mainly neutralized within lipid droplets as triglycerides, while several bioactive lipid species such as diacylglycerols (DAGs), ceramides and their derivatives have been mechanistically linked to the pathogenesis of insulin resistance (IR) by antagonizing insulin signaling and action in metabolic organs such as the liver and skeletal muscle. Skeletal muscle and the liver are the main sites of glucose disposal in the body and IR in these tissues plays a pivotal role in the development of T2D. In this review, we critically examine recent literature supporting a causal role of DAGs and ceramides in the development of IR. A particular emphasis is placed on transgenic mouse models with modulation of total DAG and ceramide pools, as well as on modulation of specific subspecies, in relation to insulin sensitivity. Collectively, although a wide number of studies converge towards the conclusion that both DAGs and ceramides cause IR in metabolic organs, there are still some uncertainties on their mechanisms of action. Recent studies reveal that subcellular localization and acyl chain composition are determinants in the biological activity of these lipotoxic lipids and should be further examined.

## 1. Introduction

The number of obese and elderly people is constantly growing in industrialized countries. Obesity and aging are independent risk factors of Type 2 diabetes (T2D) [1,2]. Thus, the number of T2D people is increasing at an alarming rate and imposes a major health and economic burden on industrialized countries [3,4]. T2D is characterized by a state of fasting hyperglycemia resulting from an impaired insulin action in metabolic organs (i.e., adipose tissue, liver and skeletal muscle) [5,6]. Insulin resistance (IR) is defined by an increased insulin requirement of the organism to maintain normal blood glucose levels [7]. Over the last few decades, a number of non-mutually exclusive hypotheses have emerged to link obesity—and age-related metabolic disorders—to IR as reviewed in detail elsewhere [8]. One prevailing hypothesis is the emergence of ectopic lipid storage in lean tissues, so-called lipotoxicity, caused by fatty acid (FA) overflow out of adipose tissue (AT) in periods of positive energy balance [9]. This phenomenon typically occurs when excess dietary lipids cannot be adequately buffered within AT stores. In fact, there is little evidence that FA per se triggers defects in insulin signaling and induces insulin resistance. Excess FA delivery and uptake in peripheral tissues such as the liver and skeletal muscle will lead to excess intracellular FA flux. These FA will first be activated to form fatty acyl-CoA and then further converted into triacylglycerols (TAG) within lipid droplets [10,11] or give rise to diacylglycerols (DAGs) and ceramides through specific metabolic pathways that will be detailed in the following chapters of this review. FA overload to the mitochondria can also cause incomplete oxidation of fatty acyl-CoA thus producing an excess of acyl-carnitine that has been associated with defects in insulin signaling, although the precise molecular mechanisms involved still remain unclear [12]. In addition, as reviewed in detail elsewhere, an inability to adequately store the excess FA influx into TAG-lipid droplets precipitate lipotoxicity [10]. An inverse association between intramyocellular TAG (IMTG) content and peripheral glucose disposal has been repeatedly reported in sedentary subjects [13,14,15]. It is now well established that IMTG is associated with elevated levels of lipotoxic intermediates such as DAGs and ceramides that inhibit insulin signaling and interfere with glucose metabolism as previously discussed elsewhere [16,17,18]. However, a growing body of data indicates that subcellular localization and composition of these bioactive lipids is a better predictor of IR than their total pool [19,20]. The purpose of this review is to summarize and discuss the most recent studies supporting a causal role of DAGs and ceramides in the development of IR, with a special attention given to transgenic mouse models influencing either the total pool or specific species and their association with insulin sensitivity.

## 2. Causal Role of DAGs in Insulin Resistance?

### 2.1. Molecular Mechanisms

During the past 20 years, DAGs have emerged as “guilty as charged” in the development of IR [17]. Early work suggested that phorbol esters, as DAG analogues, were able to alter insulin sensitivity both in vitro and in vivo [21,22,23]. This effect appeared to be mediated by Protein Kinase C (PKC) activation and translocation to the plasma membrane. Since then, novel PKC (nPKC), one of the three major sub-classes of PKC, was shown to antagonize insulin signaling. A majority of studies reported an increase of PKCθ and PKCε activation in skeletal muscle and liver in various rodent models of IR [8,24]. This association was also observed in the human muscle where both PKC demonstrated enhanced translocation in T2D subjects [25]. PKCθ has been mostly studied in skeletal muscle, while PKCε has been predominantly investigated in the context of the insulin-resistant liver [8]. Recent work supported the role of nPKC by showing that muscle-specific PKCθ knockout mice were protected from high-fat diet (HFD)-induced IR [26]. The molecular mechanisms by which PKC disrupts insulin signalling have also been deeply investigated. It was first suspected that nPKC was able to inhibit the insulin receptor (INSR) activity through phosphorylation on an unknown specific residue. In addition, even if PKCθ was described to be able to phosphorylate insulin receptor substrate (IRS1) at specific sites, no data could link a single residue modification to the insulin signaling impairment in skeletal muscle. It is believed that the large number of phosphorylation sites on INSR establishes a very complex regulation where every single residue plays a minor role [27,28]. In vitro work identified other downstream effectors of the insulin signaling pathway that were targeted by PKCs. Phosphorylation of phosphoinositide-dependent kinase 1 (PDK1) by PKCθ in muscle cells treated with palmitate was shown to decrease Protein kinase B (Akt) activating phosphorylation on Thr308 residue [29]. Another target is the guanine exchange factor GIV/Girdin which is required for normal insulin signaling. Phosphorylation of GIV by PKCθ decreases Phosphoinositide 3-kinase-Akt signalling [30]. In parallel, PKCε was found to phosphorylate INSR on Thr1160 in mouse liver, leading to a decrease of its kinase activity [31]. This residue seems to play a critical role in DAG-mediated IR. Finally, a recent phosphoproteomic study revealed that PKCε might also trigger IR through phosphorylation of ribosomal protein s6, creating a negative feedback loop on insulin signalling [32].

### 2.2. Relevant Mouse Models

Several mouse models were developed to investigate the causal link between DAG accumulation and IR. HFD is associated with increased IMTG hydrolysis reflected by elevated Adipose triglyceride lipase (ATGL) activity in mice [33]. In line with this observation, hormone-sensitive lipase knockout mice demonstrated intramuscular DAG accumulation associated with impaired insulin signalling compared to wild-type mice [33]. Similarly, liver-specific deletion of ATGL protects mice from glucose intolerance despite liver steatosis. However, HFD-fed ATGL null mice do not exhibit improved skeletal muscle insulin-stimulated glucose uptake [34]. Thus, altered lipolysis can affect the whole body and organ-specific insulin sensitivity through different mechanisms. Importantly, a number of studies suggested that only sn-1,2-DAG are able to activate PKCs [35], whereas ATGL-mediated triglyceride hydrolysis generates mainly sn-1,3-DAG [36]. Hormone-sensitive lipase knockout mice displayed increased sn-1,3-DAG accumulation after treadmill exercise along with improved insulin sensitivity compared to wild-type mice [37]. This result indicates that DAG stereo-specificity should be considered for their biological activity. In the same line, acyl chain composition and subcellular localization are other main determinants of DAG biological activity. A recent study highlighted that HFD-fed mice and endurance-trained mice display similar total DAG pools in muscle [38]. However, the quantity of DAG containing palmitoleate was reduced in HFD and increased in trained mice. An extensive study of DAG composition and localization in human muscle revealed an inverse relationship between sn-1,2-DAG in the subsarcolemmal fraction and insulin sensitivity [20]. Yet when looking at athletes, this relationship is no longer found as they present as much sn-1,2-DAG as the T2D subjects. Interestingly, the authors also reported a positive relationship between mitochondrial/endoplasmic reticulum (ER) DAG and insulin sensitivity, while the proportion of unsaturated DAG was inversely correlated to insulin sensitivity. These observations emphasize the need to produce more specific data when questioning the role of the DAG-PKC axis in the development of IR, and to consider stereo-specificity, acyl chain composition and intracellular localization.

sn-1,2-DAG are mainly produced by de novo synthesis through the Kennedy pathway [8]. Several mouse models targeting key enzymes of this pathway supported the role of DAG accumulation in the onset of IR but were also prone to a profoundly remodelled adiposity and fuel selection such as mice overexpressing lipin1 in skeletal muscle [39]. The manipulation of Diglyceride acyltransferase (DGAT) expression, which catalyzes the final conversion of DAG into TAG, yields interesting results. Overexpression of DGAT1 in skeletal muscle protected mice from HFD-induced IR [40]. Opposite findings were observed with DGAT2 overexpression [41]. The case of diacylglycerol kinase (DGK) is also very interesting as it illustrates the complexity of the mechanisms underlying the role of DAG. The enzymes of this family catalyze the conversion of DAG into phosphatic acid [42]. A decrease in DGK activity has been reported in skeletal muscle of T2D subjects associated with increased DAG content [43]. Mice with haploinsufficiency of DGKδ, an abundant isoform in skeletal muscle, display increased DAG content and impaired insulin sensitivity [43]. In contrast, whole-body deletion of DGKε triggers DAG accumulation in skeletal muscle but improves glucose tolerance [44]. Lastly, DGKζ KO mice exhibit improved muscle insulin sensitivity [45]. Altogether, these conflicting results might be explained by the specific subcellular DAG localizations and substrate preferences of the various enzyme isoforms and justify the need to further unravel the mechanisms through which DAGs drive IR. More recently, experiments in mouse embryonic fibroblasts evidenced that DAGs could accumulate through increased sphingomyelinase (SMase) activity at the plasma membrane upon palmitate treatment and deteriorate insulin signalling through PKC and c-Jun N-terminal kinases (JNK) activation [46]. This new direction seems to link both DAG and ceramides to the onset of lipid-induced IR and offers new perspectives to the field. 

## 3. Causal Role of Ceramides in Insulin Resistance?

### 3.1. Ceramides Biosynthetic Pathways

Ceramides are synthesized in all metabolic tissues in HFD-fed rodents and obese humans, and have been linked to IR as reviewed in detail in previous issues of the journal [16,47]. This appears to be independent of an increased dietary intake of ceramides, resulting rather from an increased cellular endogenous production either due to increased substrate availability or changes in synthesis enzymes [47,48,49]. Synthesis and degradation pathways and their regulation, which govern total ceramide content, have been extensively reviewed in the past few years [16,48,49]. Ceramides are mainly produced through de novo synthesis (Figure 1). The first and liming step is catalyzed by the serine palmitoyltransferase (SPT) through condensation of serine and palmitoyl-CoA into 3-ketosphinganine. The catalytic core of SPT is formed by a heterodimer of Sptlc1 and Sptlc2. Sphinganine is then generated by ketosphinganine reductase. Ceramide synthase (CerS 1-6) then performs an acylation to produce dihydroceramide. In humans, six isoforms exhibit tissue-specific expression and substrate specificity towards different acyl chain lengths [50]. Finally, dihydroceramides are converted into ceramides by dihydroceramide desaturase (DEGS 1-2). All these steps are performed at the membrane of the ER. Ceramides can transit to the Golgi apparatus through ceramide transfer protein (CERT) where they serve in the synthesis of various complex sphingolipids such as sphingomyelin (SM) and glycosphingolipids. Ceramides are also synthesized through the degradation of complex sphingolipids into sphingosine by ceramidase activity. Sphingosine is converted within the cytosol by CerS to produce ceramide through the so-called “salvage pathway”. Lastly, hydrolysis of SM by SMase also contributes to ceramide production (Figure 1).

### 3.2. Mechanistic Insights from Cell-Based Studies

Early studies investigating the effect of ceramides on insulin sensitivity in vitro used short-chain (C2, C6) cell-permeable ceramides in immortalized cell lines. A majority of these studies observed that short-term incubation with synthetic ceramides did alter insulin signaling in various cell types such as C2C12 [51] and L6 [52] myotubes, 3T3-L1 adipocytes [53,54,55] and Fao cells [56]. Exogenous ceramides seemed to interfere with distal insulin signaling at the level of Akt even though there were equivocal results (Figure 2). Some studies reported a decreased Akt phosphorylation on Ser473 and Thr308 [57,58] whereas others did not observe any effect on Thr308 [59]. The discrepancy appeared to be mediated by two distinct mechanisms. Impaired recruitment of Akt at the plasma membrane was observed, due to ceramide-activated atypical PKCζ phosphorylation of Pleckstrin homology domain on Thr34 [60,61]. Activation of protein phosphatase 2A (PP2A) was also found to directly inactivate Akt [62,63]. Studies conducted to identify the predominant mechanism of ceramide action concluded that it could depend on membrane composition. Indeed, caveolin-enriched microdomains (CEM) were necessary for ceramides to exhibit a deleterious effect on insulin signaling through PKCζ in rat aorta vascular smooth muscle cells [64] (Figure 2). On the contrary, cells with low CEM abundancy showed a predominance of PP2A-mediated mechanism. Preadipocytes devoided of CEM shifted from a PKCζ-dominant inhibition to a PP2A-dominant one [65]. C2C12 cells contain less CEM than L6 myotubes and display a PP2A-dominant inhibition while their human equivalent predominantly exhibit a PKCζ-dependent mechanism [66].

Another method to investigate the role of ceramides in vitro is to treat cells with palmitic acid (PA), which provides a substrate for both the backbone and acyl chain of the ceramide molecule. This method is thought to reproduce physiological conditions with more accuracy, as cells mostly generate long and very-long-chain ceramides. PA treatment recapitulated the effects of exogenous short ceramides in C2C12 myotubes on glycogen synthesis and Akt inhibition [51,67] but not in 3T3-L1 adipocytes [67]. 

However, the fact that most of these works only reported an impairment of insulin signaling at the level of Akt has cast doubts on the sufficiency of ceramides to account for lipid-induced IR. Indeed, most of the IR models exhibit defects in proximal insulin signaling [7]. Still, two early articles described a decreased IRS1 tyrosine phosphorylation and decreased INSR tyrosine kinase activity following treatment with ceramides [55,56]. Moreover, studies using inhibitors of ceramide synthesis demonstrated protection from PA-induced IR in C2C12 myotubes [68,69]. This protection was bypassed by the addition of C2-Cer in the culture media [70]. Chavez et al. did not observe any modification of DAG content, suggesting that sphingolipids are sufficient to induce IR in myocytes oversupplied with saturated fatty acids (SFA) [70]. Park et al. tried to identify if ceramides or complex sphingolipids were responsible for the deleterious effect on insulin signaling. They found that treatment of C2C12 myotubes did not alter SM levels and that inhibition of SM synthase activity recapitulated the effects of PA, suggesting that ceramides rather than SM were involved [71]. A recent in vitro study highlighted that CERT protein content was reduced in different lipotoxicity models and notably in PA-treated C2C12 myotubes [72]. Decreasing CERT activity through siRNA or pharmacological inhibition approaches recapitulated the effect of PA treatment, due to increased ceramide content and decreased SM. On the contrary, CERT overexpression in myoblasts decreased PA-induced ceramide accumulation, conferring protection against insulin signaling defects. This result proposes a regulatory mechanism of sphingolipid metabolism in the context of SFA oversupply and suggests once again that ceramides rather than SM are “guilty as charged”. 

Even though in vitro experiments have yielded more consistent results in muscle cells and did not seem to recapitulate all the signaling cascade defects that are observed in IR, they still offered persuasive evidence of the deleterious effects of ceramides and will continue to be a valuable tool to investigate new molecular mechanisms.

### 3.3. Mechanistic Insights from Rodent Studies

#### 3.3.1. Pharmacological Inhibition of Ceramide Synthesis

One strategy to study the impact of ceramides on insulin sensitivity in vivo is the modulation of their total pool by pharmacological inhibitors. Numerous inhibitors of the de novo synthesis pathway have been described [73]. Among the most widely used are L-cycloserine, myriocin and fumonisin B1. The first two both interfere with the first step of ceramide synthesis, though myriocin targeting of SPT appears to be much more specific. Fumonisin B1 is a potent inhibitor of CerS activity, but to the best of our knowledge, it has never been used for in vivo experiments in the context of IR. The next section will be mainly focused on the effects of myriocin in vivo in the prevention and reversal of IR.

In one seminal paper, Holland et al. demonstrated that Sprague–Dawley rats were protected against ceramide accumulation and IR when infused with lard oil emulsion and myriocin simultaneously. Myriocin also negated the deleterious effects of dexamethasone on insulin sensitivity [74]. Likewise, Yang et al. observed improved whole-body insulin and glucose tolerance in ob/ob and HFD-fed mice treated with myriocin [75]. The effect of myriocin was observed in both preventive (eight weeks of HFD combined with treatment) and curative strategies (diet was started eight weeks prior to treatment initiation). Myriocin also increased insulin-stimulated Akt phosphorylation in the liver and skeletal muscle in all of these obesity models. Myriocin drastically reduced plasma ceramide content. This was concomitant with a severe reduction in weight gain, fat mass, hepatic steatosis and a spectacular increase in locomotor activity. Myriocin also successfully restored glucose tolerance after 4 weeks of treatment in diet-induced obesity models [75]. The compound significantly limited the accumulation of ceramides in the gastrocnemius muscle but did not affect TAG, DAG and long-chain acyl-CoA content. In another study, myriocin prevented the development of IR in young db/db mice and reduced liver mass, without changes in body weight, visceral fat mass, locomotor activity and oxygen consumption [76]. More recently, Blachnio-Zabielska et al. also reported a protective effect of myriocin on insulin sensitivity in HFD-fed rats [77]. Surprisingly, the treatment also reduced DAG content in subcutaneous and visceral fat depots. In addition, myriocin was reported to protect against poly-unsaturated HFD-induced IR. In contrast with previously quoted findings, this effect was not observed with saturated HFD [78]. This result seems counter-intuitive as in lipid infusion experiments, only saturated fat administration was linked to increased ceramide content. Poly-unsaturated fat infusion also led to IR through what is believed to be a ceramide-independent mechanism, but rather DAG-dependent, as SPT inhibitors failed to prevent it [74]. However, the ceramide content of skeletal muscle did not differ after 6 weeks of both diets suggesting that the acute modulation of lipid species observed in infusion experiments might not be representative of a prolonged diet effect. The explanation for this unexpected result might simply lie in the fact that, in this particular study, myriocin reduction of ceramide content was smaller in the saturated fat-fed group. In the most detailed study employing myriocin to date, Chaurasia and colleagues demonstrated that its use in a reversal strategy (i.e., treatment for 8 weeks, 12 weeks after the transition to HFD) considerably improved insulin sensitivity in HFD-fed mice [79]. However, although myriocin reduced both serum and tissue ceramide content, HFD in this study had no visible effects on tissue ceramide content in contrast with other studies. They also showed a decrease in total fat mass associated with an elevated rate of energy expenditure and oxygen consumption without a change in food intake and locomotor activity. Fat mass loss was associated with a sharp decrease in adipocyte size in all depots. Gene expression profiling revealed a shift towards M2 macrophages along with a decrease in pro-inflammatory cytokines. All these results were confirmed in a parallel mice cohort investigating the effect of myriocin in a prevention strategy even though there was no reduction in fat mass, probably because the mice were sacrificed after 12 weeks of HFD, therefore limiting body weight gain [79]. The main discrepancies between studies with respect to phenotypic outcome could be due to changes in diet composition, treatment dose and/or diet/treatment duration.

Overall, the most interesting finding was that myriocin enhanced a thermogenic program in sWAT, but not eWAT, where ceramide levels were not reduced [79]. Thus, increased adipocyte oxidative metabolism and thermogenesis, i.e., the so-called browning process of white fat, could be another mechanism by which myriocin improves metabolic status in obese mice. This direction will be discussed in light of the data that emerged from transgenic models in the next chapter. Collectively, preclinical studies using pharmacological inhibitors of ceramide synthesis such as myriocin largely support a preventive and curative effect in obesity-related IR and T2D development.

#### 3.3.2. Transgenic Mouse Models Modulating Total Ceramide Pool

Studies in transgenic mice facilitated a breakthrough in the field and a better understanding of the causative role of ceramides in T2D development. A comprehensive list of seminal studies is presented in Table 1. Li et al. generated a model of mice with a heterozygous whole-body deletion of Sptlc2 [80]. Unexpectedly, these mice under HFD paradoxically exhibited increased liver ceramides while no change in muscle and adipose tissue (AT) when compared to wild-type controls. Only plasma levels of sphingolipids significantly dropped, alongside TAG and FFA. This was associated with an improvement of insulin sensitivity as measured by glucose and insulin tolerance tests, as well as an increased INSR/Akt phosphorylation in muscle, liver and AT. All these improvements, although more subtle, were also observed in chow-fed mice. Reduced weight and fat mass gain were reported in HFD-fed heterozygous mice. In a later study, AT-specific knockdown of Sptlc2 improved whole-body insulin sensitivity in HFD-fed mice and to a lesser extent in chow-fed mice [79]. However, the authors observed a significant decrease in ceramide and/or dihydroceramide content in all fat pads investigated. This effect was markedly greater for very-long-chain ceramides but lower C16:0 and C18:0 ceramide levels were detected in sWAT primary adipocytes. Once again, these mice gained less weight than WT littermates on an HFD with markedly reduced fat mass and smaller adipocytes. This could be explained by greater energy dissipation through non-shivering thermogenesis as reflected by an elevated rectal temperature. Although these results corroborate the findings of Li et al. [80], they did not dispel doubt on the causal role of ceramides in lipid-induced IR. Indeed, modulating the de novo ceramide synthesis pathway tends to influence the content of numerous sphingolipid species. 

In addition, two studies investigated the effect of Degs1 deficiency on insulin sensitivity [75,81]. As stated above, Degs1 is responsible for the desaturation of dihydroceramide into active ceramides. Expectedly, homozygous null mice exhibited very few ceramides while dihydroceramide levels were extremely high [74]. However, these animals displayed significant prenatal lethality and for the survivors, a myriad of abnormalities leading to premature death. In contrast, heterozygous Degs1^+/−^ mice were healthy and their ceramide/dihydroceramide ratio was decreased in various tissues including WAT, *soleus* muscle and liver. These mice exhibited improved insulin sensitivity and were protected from dexamethasone-induced IR. Chaurasia and colleagues recently proposed a deeper analysis of Degs1 deficiency by crossing *ob/ob* mice with Degs1^Rosa26/ERT2-Cre^ mice [81]. Upon tamoxifen administration, excision of Degs1 induced a severe reduction of ceramide/dihydroceramide ratios in the liver, WAT and skeletal muscle, for all the acyl chain length tested. Although this was performed on a genetically obese background, Degs1 ablation prevented age-induced weight gain compared to control littermates. These mice had reduced fat mass and displayed smaller adipocytes in eWAT and sWAT. This phenotype appeared independent of ambulatory activity and changes in food intake. Of interest, their respiratory exchange ratio was slightly decreased which reflects an elevated rate of lipid oxidation. Ablation of Degs1 in ob/ob mice also reversed hepatic steatosis and ameliorated whole-body insulin sensitivity. Metabolic phenotyping was performed shortly after tamoxifen administration (i.e., 3 and 4 weeks), while mice presented similar body weight. To gain further insight into the contribution of specific tissues, the same investigators induced a selective deletion of Degs1 in the liver and AT [81]. In both mouse models, the ceramide/dihydroceramide ratio diminished and insulin sensitivity was augmented under HFD. This effect was already significant after 4 weeks of HFD even though weight gain was moderate and comparable between WT and KO mice. Targeted ablation of Degs1 in AT and liver decreased liver fat independently of changes in body weight. These findings were further confirmed by the use of shRNA targeting Degs1 transcripts [81].

#### 3.3.3. Transgenic Mouse Models Modulating Specific Ceramides Species

Transgenic mice harboring modulation of ceramide synthase (CerS) expression provided insightful data with respect to the causal link between ceramides and IR. Six isoforms of CerS were described and each demonstrates a preference for acyl-CoA of a specific length. CerS6 possesses a high affinity towards C16:0 acyl-CoA. Whole-body deletion of CerS6 led to a reduction of C16:0 ceramide in WAT, BAT and liver of mice submitted to a 17-week HFD [82]. This decrease was not observed in skeletal muscle where CerS6 expression is very low and C16:0 ceramide content negligible compared to other metabolic tissues. Overall, these mice were protected from diet-induced IR as revealed by insulin and glucose tolerance tests but also by increased phosphorylation of Akt and Glycogen synthase kinase 3β in the liver after insulin injection, but not in skeletal muscle. Body weight and fat mass gain were strikingly reduced, with decreased adipocyte size in WAT. Once again, the authors reported an elevated rate of energy expenditure, which could possibly explain the resistance to HFD-induced obesity. Insulin sensitivity was also enhanced in chow-diet Cers6^−/−^ mice. They also concluded that BAT function was improved due to increased lipid oxidation capacity in this tissue. Specific CerS6 deletion in BAT yielded the same observations. Cers6^ΔBAT^ HFD-fed mice displayed improved glucose tolerance, increased energy expenditure and reduced adiposity even if differences in body weight were less pronounced compared to whole-body deletion. Liver-specific CerS6 deletion also improved glucose tolerance and modestly reduced weight gain in HFD-fed mice. These data suggest that C16:0 ceramides or related complex sphingolipids are detrimental for insulin sensitivity in AT and liver. Similar observations were made by Raichur et al. with the use of antisense oligonucleotides (ASO) directed against CerS6 transcripts [83]. As C16:0 ceramides were decreased in plasma and liver of ASO treated ob/ob mice compared to control mice, they noted a compensatory increase in very-long-chain ceramides. However, they confirmed that C16:0 ceramide depletion improved metabolic status. This effect was once again accompanied by decreased weight gain and food intake. A 6-week ASO treatment in HFD-fed mice triggered a continuous weight and fat mass loss, but food intake was not different between treated and untreated mice. As C16:0 is also generated by the CerS5 isoform, Hammerschmidt and colleagues compared the effects of whole-body CerS5 and CerS6 deletions [84]. In CerS6^−/−^ mice, liver expressions of CerS2, CerS4 and CerS5 were slightly enhanced. That was not the case for CerS5 deletion. Additionally, liver C16:0 ceramides were more decreased in CerS6^−/−^ mice (60%) than in CerS5^−/−^ mice (40%). Only CerS6 deletion led to a decrease in weight gain and adiposity in HFD-fed mice. These mice were protected from diet-induced IR whereas CerS5 KO mice were not, exhibiting even more degradation than WT mice during an ITT. The authors went on to investigate the modifications in ceramide content at a subcellular scale. Fractionation experiments on liver homogenates revealed that HFD-fed mice exhibit elevated expression of CerS6 but not CerS5. C16:0 ceramides were found to increase in the mitochondrial and mitochondrial-associated membrane (MAM) fractions of these animals. Surprisingly, only CerS6-deleted mice displayed a reduction in mitochondria and MAM C16:0 ceramides. In contrast, CerS5^−/−^ mice displayed increased CerS6 expression in these fractions and no significant difference in C16:0 ceramide content. While it is not entirely clear whether the specific subcellular localization of CerS isoforms and their respective byproducts triggered phenotypic differences between CerS5^−/−^ and CerS6^−/−^, this highlights the fact that both chain length and localization of ceramides are determinant for IR.

Evidence of the deleterious effects of shorter acyl-chain ceramides has also been observed in skeletal muscle. Unlike in the liver and AT, C16:0 ceramide content is very low and C18:0 ceramides are abundant due to predominant CerS1 expression. Whole-body deletion of CerS1 seems to predominantly affect the sphingolipid content in the skeletal muscle of HFD-fed mice [85]. Indeed, CerS1 ablation did not impact the expression of other CerS isoforms. CerS1 deletion triggered a significant reduction of C18:0 ceramides in the quadriceps muscle, and a concomitant increase in other ceramide subspecies, including C16:0. CerS1^−/−^ mice gained less weight upon HFD, similarly to CerS6^−/−^ mice despite slightly elevated food intake. Once again, the authors noticed a reduction in fat mass and increased energy expenditure. These mice were protected from diet-induced IR. However, CerS1^−/−^ mice displayed various abnormalities such as cerebellar ataxia that could strongly interfere with the measured metabolic outcome. Skeletal muscle-specific CerS1 KO recapitulated the positive effect on insulin sensitivity in HFD-fed mice [85]. However, these mice were not protected from weight gain and displayed similar energy expenditure compared to their relative controls in contrast with CerS1 KO. This could be due to improved insulin-stimulated glucose uptake in the skeletal muscle, and also to increased suppression of hepatic glucose production. This was not due to a decrease in C18:0 ceramide content in the liver. Unexpectedly, there was no difference in insulin-stimulated Akt phosphorylation or PP2A activity in the quadriceps muscle. Mitochondrial content, morphology and respiratory activity were not affected either. Turner et al. selectively inhibited CerS1 activity in HFD-fed mice using a pharmacological derivative of fingolimod [86]. This inhibition led to a decrease in C18:0 ceramides and related SM and a parallel increase in very-long-chain ceramides in the skeletal muscle, as well as a markedly reduced TAG content. Ceramide content was not affected in the liver and AT. Mice treated with the inhibitor gained less weight and displayed smaller WAT depots but were not protected from diet-induced IR. This result is even more surprising considering that unlike Turpin-Nolan et al. [85], the authors observed an increase of mitochondrial respiratory activity and lipid oxidation in skeletal muscle. 

**Table 1 ijms-21-06358-t001:** Transgenic mouse models of whole-body and tissue-specific modulation of total and species-specific ceramide and sphingolipid pools.

Mouse Models	Diet/Treatment	BW/BC	Plasma Changes	Tissue Changes	Readouts of Insulin Sensitivity	Readouts of Insulin Signaling	References
*Degs1^+/−^*	CD (7w)	-	-	**SkM** ↓ Cer/DhCer **WAT** ↓ Cer/DhCer **Liver** ↓ Cer/DhCer	↑ Insulin sensitivity index ↑ Insulin tolerance	-	Holland et al., 2007 [74]
	CD + Dexamethasone (6w + 1w)	-	-		↑↑ Insulin sensitivity index ↑↑ Insulin tolerance	-	
*Sptlc2^+/−^*	CD (16w)	↔ BW	↔	**SkM** ↔ **AT** ↔ **Liver** ↑ Cer	↑ Glucose tolerance ↑ Insulin tolerance ↔ Pyruvate tolerance	**AT + Liver** ↑↑ INSR phosphorylation ↔ Total INSR ↑↑ Akt p-S473	Li et al., 2011 [80]
	HFD (16w)	↓ BW gain	↓ SM S1P ↓↓ Cer	**SkM** ↔ **AT** ↔ Sphingolipids ↓ Adipocyte size **Liver** ↑ Cer		**SkM + AT + Liver** ↑↑ INSR phosphorylation ↔ Total INSR ↑↑ Akt p-S473 **SkM + AT** ↑ GLUT4 translocation	
*Sms2^+/−^*	CD (16w)	↔ BW	-	**SkM** ↓ SM ↑↑ Cer **AT** ↓ SM **Liver** ↓ SM ↑↑ Cer	↑ Glucose tolerance ↑ Insulin tolerance **SkM + WAT** ↑ glucose uptake	Liver ↑↑ Akt p-S473	
	HFD (16w)	↓ BW gain	-	-	↑ Glucose tolerance ↑ Insulin tolerance ↔ Pyruvate tolerance	-	
*Cerk^−/−^*	HFD (11w)	↓ BW gain	-	**Brain** ↔ C1P **AT** ↓↓ Adipocyte size ↓↓ Inflammation	↑ Glucose tolerance	eWAT ↑↑ Glut4 + INSR expression	Mitsutake et al., 2012 [87]
CerS2^−/−^	CD	↓ BW ↔ BC	-	**Liver** ↓↓ C22, C24 Cer ↑↑ C16 Cer, HexCer, SM ↑↑ Sphinganine ↑↑ nSMase activity ↓ glycogen storage	↑ Fasting/Fed plasma glucose ↓ Insulin tolerance ↓ Glucose tolerance	**Liver** ↓↓ p-INSR, Akt p-S473 **SkM/AT** ↔	Park et al., 2013 [88]
CerS2^+/−^	CD (12w)	↔ BW gain ↓ BW/Lean mass	-	**Liver** ↓↓ C22:0/C24:0 Cer ↑↑ C16:0 GlcCer ↑↑ C16:0 DhSM ↑↑ C16:0 ↓↓ C24:0 DhCer	↔ Plasma Insulin ↔ Glucose tolerance ↔ Insulin tolerance	-	Raichur et al., 2014 [89]
	HFD (12w)	↔ BW gain ↑↑ Fat mass/Lean mass	-	**Liver** ↑ weight ↑ TAG ↑ macrophages ↑↑ C16:0, C18:0, C24:1 Cer ↑ Total Cer ↓↓ Total DhCer ↑ C16:0, C18:1, C24:1 SM ↓↓ C26:0 SM ↑↑ Sphinganine, Sphingosine ↑↑ C16:0 GlcCer ↓↓ C22:0, C24:0 GlcCer	↑ Fed and Fasted plasma Insulin ↓ Glucose tolerance ↓ Insulin tolerance	-	
CerS6^−/−^	CD (17w)	↓ BW gain	-	-	↑ Glucose tolerance ↔ Insulin tolerance	-	Turpin et al., 2014 [82]
	HFD (17w)	↓ BW gain ↓ Fat mass	-	**WAT** ↓↓ Adipocyte size ↓↓ MAC2 positive cells ↓↓ C16:0 Cer ↑ C18:0 Cer **BAT** ↓↓ C16:0 Cer ↓↓ LD size ↑↑ Lipolysis, ↑↑ β-oxydation capacity **Liver** ↓↓ C16:0 Cer SkM ↔ Cer	↓ Serum insulin ↑ Glucose tolerance ↑ Insulin tolerance	**Liver** ↑↑ Akt p-T308, p-S473 ↑↑ p-GSK3β **SkM** ↔	
Cers6^ΔBAT^	HFD (17w)	↔ BW gain ↓ Fat mass	-	**BAT** ↑↑ β-oxydation capacity	↑ Glucose tolerance ↔ Insulin tolerance	-	
Cers6^ΔLiver^	HFD (17w)	↓ BW gain	-	**Liver** ↓↓ C16:0 Cer, DhCer	↑ Glucose tolerance ↔ Insulin tolerance	-	
Liver-specific overexpression of AC	HFD + Dox (8w)	↔ BW gain	↓↓ C16:0, C18:0, Total Cer ↓↓ Sphingosine Sphinganine	**Liver** ↓ weight ↓↓ C16:0, C18:0, C20:0 Cer ↓ Sphingosine ↓↓ TAG ↑↑ DAG ↓ FA synthesis and uptake gene expression **AT** ↓ GlcCer, DhCer, LacCer ↓↓ Sphingosine Sphinganine Fat pad weight redistribution ↓ inflammation	↑ Glucose tolerance ↑ Insulin tolerance ↑ HE clamp **gWAT + sWAT + mWAT** ↑↑ Glucose uptake	**Liver + gWAT** ↑↑ Akt p-S473	Xia et al., 2015 [90]
	CD + Dox (8w)	-	-	-	↑ Glucose tolerance	-	
	HFD − HFD + Dox (8w–8w)	-	-	**Liver** ↑↑ TAG ↓↓ C16:0 C18:0 Cer	↔ Insulin tolerance (3d post-induction)	-	
AT-specific overexpression of AC	HFD + Dox (8w)	↔ BW gain	↓ Cer GluCer ↑↑ Sphingosine Sphinganine ↑↑ S1P	**mWAT+ gWAT + sWAT** ↓ weight (gWAT) ↓↓ C16:0, C18:0, Total Cer ↓ Inflammation **Liver** ↓↓ TAG ↓↓ C16:0 C18:0 Cer	↑ Glucose tolerance ↑ Insulin tolerance ↑ HE clamp **gWAT + sWAT + mWAT** ↑↑ Glucose uptake	**Liver + gWAT** ↑↑ Akt p-S473	Xia et al., 2015 [90]
	CD + Dox (8w)	-	-	**mWAT** ↓ C18:0, C24:0, C24:1 Total Cer **Liver** ↓ Total Cer ↓↓ TAG	↑ Glucose tolerance	-	
	HFD − HFD + Dox (8w–8w)	-	-	**Liver** ↓↓ C16:0, C18:0 Cer ↓↓ TAG	↑ Insulin tolerance (3d post-induction)	-	
CerS5^−/−^	CD (16w)	↔ BW ↔ eWAT mass	↑↑ C24;0 Cer, SM ↓ S1P	**SkM** ↓ C16:0 Cer **Liver** ↓↓ C16:0 SM **eWAT** ↔	↔ Glucose tolerance ↔ Insulin tolerance	**eWAT** ↑ Akt p-S473	Gosejacob et al., 2016 [91]
	HFD (16w)	↓ BW gain ↓ eWAT mass	↓ C16:0, C20:0 SM ↓ S1P	**SkM** ↓↓ C16:0, C18:0 Cer, SM **Liver** ↓↓ C16:0 Cer, SM ↑ C18:0 SM **eWAT** ↓↓ C16:0, C18:0 Cer ↓↓ Adipocyte size ↓ proinflammatory cytokines	↑ Glucose tolerance ↑ Insulin tolerance	-	
Sms2^−/−^	CD (12–23w)	↔ BW	↓ C22:0, 24:0 SM ↑↑ C20:0-C24:0 Cer ↑↑ C22:0 HexCer	**Liver** ↓↓ C20:0-C24:0 SM ↑↑ C20–24:0 HexCer **SkM** Very few modifications in SM or Cer content ↔ TAG, DAG	↔ Fasting plasma insulin ↔ Fasting plasma glucose ↔ HOMA-IR ↑ Glucose tolerance ↑ Insulin tolerance ↑ 18F-FDG clearance	**Liver/SkM** ↑↑ Akt p-S473 **mWAT** ↓↓ Akt p-S473	Sugimoto et al., 2016 [92]
	HFD (12–23w)	↓ BW gain	↓ C16:0-C24:0 SM ↑↑ C20:0-C24:0 Cer ↑↑ C22:0 HexCer	**Liver** ↓ C18:0-C22:0 SM ↑↑ C22:0 Cer ↓↓ C24:1 DhCer ↑↑ C16:0-C22:0 HexCer **SkM** ↓ TAG, DAG ↔ NEFA	↓ Fasting plasma insulin ↔ Fasting plasma glucose ↓ HOMA-IR ↑ Glucose tolerance ↑ Insulin tolerance ↑ ^18^F-FDG clearance **SkM + Liver + Heart** ↑ ^18^F-FDG uptake	**Liver/SkM** ↑↑ p-INSR, Akt p-S473 **mWAT** ↑↑ Akt p-S473	
Sms2^ΔLiver^	HFD (32w)	↔ BW	↓ C20:0–24:0 SM ↑ C22:0, 24:0, 24:1 Cer ↑ HexCer	**Liver** ↓ C20:0–24:0 SM ↑↑ HexCer **SkM** ↔	↔ Fasting plasma glucose ↔ Fasting plasma insulin ↔ Glucose tolerance ↔ Insulin tolerance	-	
Sptlc2^ΔAdipo^	CD (12–16w)	-	↔ Cer, DhCer, DHC, GM3	**Primary Adipocytes** **eWAT** ↓ C22:0, C24:0, C24:1, Total Cer **sWAT** ↓↓ All Cer **BAT** ↓↓ C22:0, C24:0, C24:1, Total Cer **AT** **eWAT** ↓ Total Cer, ↓↓ Adipocyte size **sWAT** ↓ C16:0, C24:0, Total Cer ↓↓ DhCer, MHCer, ↓↓ Adipocyte size ↑↑ Thermogenic gene expression, ↑ Uncoupled respiration and OCR **BAT** ↓↓ C24:0, C24:1, Total Cer	↔ Fed/Fasting plasma glucose ↔ HOMA-IR ↑ Glucose tolerance	-	Chaurasia et al., 2016 [79]
	HFD (12–16w)	↓ BW gain ↓ Fat mass/Lean mass ratio	-	**BAT** ↑ Basal respiration **Liver** ↓ Lipid storage	↓ Fed/Fasting plasma glucose ↑ HOMA-IR ↑ Glucose tolerance ↑ HE clamp Glucose uptake **sWAT** ↑ **eWAT + BAT** ↑↑ **SkM** ↔	-	
CerS6 KD (ASO)	HFD (18w + 6w ASO)	↓ BW and Fat mass	-	**Liver** ↓ TAG ↑↑ Glycogen ↑ Gys2 mRNA	↔ Fed blood glucose ↓ Fed/Fasting plasma Insulin ↓ HOMA-IR ↔ Glucose tolerance ↑ Insulin tolerance	-	Raichur et al., 2019 [83]
	CD (ob/ob)	↓ BW gain ↓ Fat mass (compared to pre-treatment)	↓↓ C16:0 Cer ↑↑ C18:0, C20:0 Cer ↑ C22:0, C24:0, C24:1 Cer	**Liver** ↓↓ C16:0 Cer ↑↑ C20:0, C22:0, C24:0, C24:1 Cer	↓ Fasted plasma glucose ↓ Fasted plasma insulin ↑ HOMA-IR ↑ Glucose tolerance ↑ Insulin tolerance	-	
CerS1^−/−^	HFD (17w)	↓ BW gain ↓ Fat mass	-	**SkM** ↓↓ C18:0 Cer, DhCer, SM ↑↑ C16:0, C22:0, C24:0, C24:1 Cer, DhCer, SM ↔ DAG TAG **Heart + Liver + WAT** ↔ Cer, DhCer, SM	↑ Glucose tolerance ↑ Insulin tolerance	-	Turpin-Nolan et al., 2019 [85]
CerS1^ΔSkM^	HFD (17w)	↔ BW gain ↔ Fat mass	-	**SkM** ↓↓ C18:0 Cer, DhCer ↑↑ C16:0 Cer ↑↑ C22:1, C24:0, C24:1 Cer, DhCer, SM ↓↓ C22:0 Cer ↔ DAG TAG **Heart + Liver + WAT** ↔ Cer	↑ Glucose tolerance ↑ Insulin tolerance ↑ HE Clamp (↑ suppression of hepatic glucose production) Glucose uptake **SkM** ↑ (Trend)	↔ Akt p-T308, p-S473 ↔ PP2A activity	
CerS5^ΔSkM^ + CerS6^ΔSkM^	HFD (17w)	↔ BW gain ↔ Fat mass	-	**SkM + Heart + Liver+ WAT** ↔ Cer, DhCer, SM	↔ Glucose tolerance ↔ Insulin tolerance	-	
ob/ob Degs1^Rosa26/ERT2-Cre^	CD (12w)	↓ BW gain ↓ Fat mass	↓↓ Cer/DhCer, Cer ↓↓ SM, Sphingosine	**Liver + WAT + SkM** ↓↓ Cer/DhCer ↑↑ DhCer **eWAT + sWAT** ↓ Adipocyte size ↓ SM ↔ Cer Liver ↓↓ LD area	↑ Glucose tolerance ↑ Insulin tolerance	-	Chaurassia et al., 2019 [81]
Degs1^ΔAdipo^	HFD (12w)	↔ BW	↔ Cer/DhCer	**eWAT + sWAT+ BAT + SkM** ↓↓ Cer/DhCer ↔ Cer **Liver** ↔ Cer/DhCer, Cer **Liver + SkM** ↓↓ SM	↓ Fed/Fasting plasma glucose ↓ Serum insulin ↑ Glucose tolerance ↑ Insulin tolerance	-	
Degs1^ΔLiver^	HFD (12w)	↔ BW	↓↓ Cer/DhCer ↓↓ Cer ↑↑ DhCer ↓↓ SM	**Liver + SkM** ↓↓ Cer/DhCer **SkM** ↓↓ SM	↔ Fed/Fasting plasma glucose ↓ Serum insulin ↑ Glucose tolerance ↑ Insulin tolerance	-	
Degs1^ΔLiver/Adipo^	HFD (12w)	↔ BW	↓↓ Cer/DhCer ↓↓ Cer ↑↑ DhCer ↓↓ SM	**BAT + Liver + eWAT + sWAT + SkM** ↓ Cer/DhCer ↓↓ Cer ↑↑ DhCer **Liver + SkM** ↓↓ SM	↓ Fed/Fasting plasma glucose ↓ Serum insulin ↑ Glucose tolerance ↑ Insulin tolerance	-	
CerS5^−/−^	HFD (13–17w)	↔ BW ↔ Fat mass	-	**Liver** ↓↓ C16:0, C22:0, C24:0 Cer	↔ Glucose tolerance ↓ Insulin tolerance	-	Hammerschmidt et al., 2019 [84]
CerS6^−/−^	HFD (13–17w)	↓ BW gain ↓↓ Fat mass	-	**Liver** ↓↓ C16:0 ↓ C16:1	↑ Glucose tolerance ↑ Insulin tolerance	-	
CerS6^iKO^	HFD (13–17w)	↓ BW gain	-	**Liver** ↓↓ C16:0	↑ Glucose tolerance ↑ Insulin tolerance	-	
CerS6^fl/fl^ + AAV8-TBG-iCre	HFD (13–17w)	↔ BW ↔ Fat mass	-	**Liver** ↓ C16:0	↑ Glucose tolerance ↔ Insulin tolerance ↔ Pyruvate tolerance	-	
WT + AAV8-TBG-CerS6	HFD (13–17w)	↔ BW-	-	**Liver** ↑↑ C16:0	↔ Glucose tolerance ↔ Insulin tolerance ↓ Pyruvate tolerance	-	

Table legend: ↑ increase; ↑↑ large increase; ↓ decrease; ↓↓ large decrease; ↔ no change; AC acid ceramidase; Akt protein kinase B; AT adipose tissue; BAT brown AT; BC body composition; BW bodyweight; CD chow diet; Cer Ceramide; d day; DAG diacylglycerol; DhCer dihydroceramide; Dox doxycycline; GLUT4 glucose transporter type 4; GlcCer glucosylceramide; GSK3β glycogen synthase kinase 3β; Gys2 Glycogen synthase 2; HE clamp hyperinsulinemic euglycemic clamp; HexCer hexosylceramide; HFD high-fat diet; HOMA-IR homeostatic model assessment for insulin resistance; INSR insulin receptor; LacCer lactosylceramide; LD lipid droplet; MHCer monohexosylceramide; NEFA non esterified fatty acids; PP2A protein phosphatase 2A; S1P sphingosine-1-phosphate; SkM skeletal muscle; SM sphingomyelin; TAG triacylglycerol; w week; WAT white AT.

CerS2 is an isoform responsible for the generation of very-long-chain ceramides (C22:0/C24:0/C24:1). Homozygous deletion of CerS2 was found to alter whole-body and liver insulin sensitivity in chow-fed mice [88]. Despite being lighter than their WT littermates and displaying the same body composition, insulin-stimulated IR and Akt phosphorylation were significantly reduced in the liver, but not skeletal muscle or AT. As expected, these mice had decreased hepatic very-long-chain ceramides but markedly increased C16:0 ceramides. Interestingly, SMase activity was also enhanced. However, these mice developed hepatocellular carcinomas and had a shortened lifespan of only 16 months. To bypass this limitation, Raichur and colleagues generated mice with a heterozygous deletion of CerS2 [89]. They demonstrated a modest decrease in very-long-chain sphingolipids in the liver, but more importantly, a sharp increase in C16:0 ceramides. When submitted to an HFD, they displayed an increased fat-to-lean mass ratio as well as liver weight and TAG content. This was independent of changes in respiratory exchange ratio and oxygen consumption. However, heterozygous mice displayed reduced locomotor activity. They were more susceptible to diet-induced IR and their hepatocytes showed reduced fatty acid oxidation rates. These data confirmed that shorter chain ceramides and related sphingolipids have a causal role in the onset of IR and that the increase of C16:0/C18:0-containing sphingolipids but not the decrease in very-long-chain, long-chain-containing sphingolipids drives IR. In summary, data from mouse models manipulating the content of C16:0 and C18:0-containing sphingolipids coincide and demonstrate their deleterious impact in all insulin-sensitive tissues.

#### 3.3.4. Ceramides or Complex Sphingolipids as Culprit of IR?

In various transgenic mouse models, it is still unclear whether IR results from changes in ceramides pools and subspecies or also from changes in complex sphingolipids such as SM and gangliosides. Indeed, both in vitro and in vivo data have incriminated SM and GM3 ganglioside [93,94,95,96].

For instance, Li et al. investigated the effect of whole-body Sphingomyelin synthase 2 (Sms2) heterozygous deletion [80]. Chow-fed mice exhibited reduced SM content in the plasma, liver, muscle and AT, while total ceramide content increased in skeletal muscle and liver. Despite ceramides accumulation, both chow-fed and HFD-fed mice had improved insulin sensitivity. Insulin-stimulated glucose uptake was increased in muscle and AT, while insulin-mediated Akt phosphorylation was improved in the liver. In line with previous observations, Sms2-deficient mice gained less weight upon HFD reflected by a sharp reduction in adipocyte size. The authors hypothesized that this marked AT phenotype could be due to the reduction of SM content at the plasma membrane of adipocytes. In addition, only the total ceramide content was quantified. A more extensive study of the effect of Sms2 deletion supported these conclusions. Homozygous Sms2 KO mice were protected against HFD-induced IR and gained less weight than their WT counterparts [92]. Additionally, they displayed overall decreased SM levels and increased very-long-chain ceramides, but not shorter chain ceramides, in the liver and plasma. No major differences in sphingolipid content were observed in the skeletal muscle, but TAG and DAG content was reduced in HFD-fed animals. Skeletal muscle glucose uptake was enhanced in KO mice even though the mechanism behind this improvement was not deeply investigated. In contrast to Li et al., Sms2 heterozygous deletion did not affect whole-body insulin sensitivity nor body weight upon HFD administration. Liver-specific Sms2 deletion did not improve insulin sensitivity either [92].

In addition, relevant information has arisen from studies manipulating enzymes responsible for the degradation/transformation of ceramides. Overexpression of acid ceramidase (AC) in the liver of HFD-fed mice led to a robust decrease of liver and serum C16:0/C18:0 ceramides [90]. Despite a significant increase in liver DAG content and no difference in body weight compared to WT, these mice were less susceptible to diet-induced IR as evidenced by hyperinsulinemic–euglycemic clamps studies and improved insulin-stimulated phosphorylation of Akt in the liver. Improved glucose disposal was not limited to the liver and was observed in all WAT fat pads. AC overexpression reversed diet-induced liver steatosis through increased Very Low Density Lipoprotein secretion and reduced liver lipid uptake without apparent changes in lipid oxidation. Very similar outcomes were reported when the overexpression of AC was performed, specifically in AT. Strikingly, IR and liver steatosis were fully rescued in these mice while this effect was much more delayed in mice overexpressing AC specifically in liver under HFD. These data support a major causal role of ceramides in lipid-induced IR.

Mouse models manipulating CerS activity indicated that any shift in ceramide subspecies can impact metabolic regulation and insulin sensitivity regardless of the total ceramide pool. Thus, specific ceramide species appear to be implicated in the onset of IR in all metabolic tissues (i.e., skeletal muscle, liver, WAT and BAT). In most cases, improving the sphingolipid profile in one of these tissues contributes to alleviating whole-body IR through tissue crosstalk. This suggests that ceramides might be exchanged between metabolic tissues and reveal unsuspected crosstalk between liver and AT. In addition, manipulating tissue-specific ceramides pools could influence other parameters such as secretory function indirectly mediating the observed phenotypes. As previously discussed, depleting short-chain ceramides in AT greatly reduces liver steatosis and improves liver insulin sensitivity [90]. Moreover, targeting whole-body ceramide pools seems to affect metabolism to a higher extent than organ-specific modulations.

Still, it is not possible to rule out the role of other complex sphingolipids in lipid-induced IR. Two studies cited previously reported a positive effect of SM activity inhibition, suggesting a causal role of SM in IR [80,92]. It is also possible that complex sphingolipids are further degraded by SMase or in the salvage pathway and contribute to the remodeling of the acyl-chain and subcellular localization profile of the ceramide pool. Future investigations will have to discriminate between these main suspects. Generating new mouse models that manipulate SMase or ceramidase activities in a tissue-specific manner will provide new perspectives. Another concern is to understand the association between ceramide subcellular localization and their lipotoxic effects. Recent evidence suggests that sarcolemmal and mitochondrial C18 ceramides and SM are inversely related to insulin sensitivity whereas cytosolic sphingolipids are not in the human skeletal muscle [20]. CerS6^−/−^ but not CerS5^−/−^ mice displayed reduced mitochondrial-associated C16:0 ceramides in the liver, and this was related to improved whole-body insulin sensitivity [84]. This finding highlights the urgent need to further decipher the distinct cellular effects of ceramides depending on their subcellular localization and identify by which mechanisms they are compartmentalized within the cell in the context of obesity and aging.

### 3.4. Emerging Mechanisms of Action of Ceramides

As previously discussed, in vitro studies provided mechanistic insights into the molecular mechanisms by which ceramides disrupt insulin signaling. Molecular alterations in distal signaling were observed at the level of Akt phosphorylation, mostly mediated by ceramide-activated PP2A and PKCζ. However, defects in proximal signaling have often been reported in this context and more recently incriminated ceramides. Prolonged incubation of C2C12 myotubes with PA provoked increased IRS1 Ser307 phosphorylation [97]. This effect appeared to be mediated by double-stranded RNA protein kinase (PKR) through JNK activation. This finding highlights the fact that short-term ceramide toxicity in vitro might not reflect the broad effects that are observed in vivo and therefore fail to reveal all underlying mechanisms. It is also striking that most of the mouse models reducing total ceramide pools or manipulating ceramides acyl chain composition report protection against HFD-induced weight gain. This has been frequently seen as a confounding factor in in vivo models challenging the causal role of ceramides in the development of IR. This particular phenotypic feature appears fully driven by whole-body and tissue ceramide/sphingolipid depletion. This appears also independent of reduced food intake or increased locomotor activity in most models, thus suggesting that resistance to weight gain is triggered by peripheral mechanisms increasing energy expenditure.

In this regard, Chaurasia et al. noticed the induction of a brown/beige fat thermogenic program in sWAT but not in eWAT or BAT of HFD-fed mice treated with myriocin [79]. This was confirmed by increased Uncoupling Protein 1, T-box transcription factor 1 and Peroxisome proliferator-activated receptor gamma coactivator 1-alpha protein levels. They observed a similar effect of adipose-specific Sptlc2 deletion. This was associated with increased uncoupled respiration and mitochondrial complex activity in sWAT. Lipid oxidation was higher in sWAT and BAT. They went on to show that cold exposure or treatment with a β-adrenergic agonist both decreased ceramides and related intermediates in sWAT and eWAT. Furthermore, β-adrenergic agonist administration triggered elevated energy expenditure in Sptlc2^δAdipo^ mice compared to WT animals. They also recapitulated the deleterious effect of C2-ceramides on mitochondrial respiration in cultured sWAT primary adipocytes [79]. This work provided compelling evidence of the causal role of ceramides in the alteration of mitochondrial activity in WAT adipocytes. In a recent study, it was demonstrated that ceramide depletion, through Degs1 deletion, improved the activity of mitochondrial complexes I, II and IV [81]. Modulation of the total pool of ceramides does not seem to be the sole factor influencing mitochondrial respiratory activity. Several studies describe a specific role of C16:0- and C18:0-containing ceramides. Chronic pharmacological inhibition of CerS1 resulted in increased PA oxidation in *soleus* muscle [86]. Analysis of mitochondrial function revealed an increase in mitochondrial capacity reflected by the greater expression of mitochondrial complexes and activity specifically in skeletal muscle where C18:0 ceramides are the most abundant. CerS1 inhibition also promoted mitochondrial biogenesis traduced by increased PGC1α protein level and mitochondrial DNA content. Consistent with these data, hepatocytes from mice with CerS2 haploinsufficiency exhibited decreased basal respiration, ATP turnover as well as maximal and spare respiratory capacity [89]. This was associated with decreased complex 2 and 4 activity, which could be reversed, at least for complex 4, by a 2-week myriocin treatment independent of mitochondrial content. Overexpression of CerS6 in isolated primary hepatocytes inhibited complex 2 activity [83]. In contrast, primary hepatocytes from CerS6 deleted mice displayed greater rates of PA oxidation. These results evoke that C16:0 ceramides cause mitochondrial dysfunction. CerS6, but not CerS5 deletion, improved ADP-stimulated respiration of isolated hepatic mitochondria [84]. The authors observed fragmentation of the mitochondrial network independent of changes in mitochondrial content in the liver of WT obese mice. As stated before, a reduction in mitochondria and mitochondrial-associated membrane (MAM) C16:0 ceramides was observed specifically in CerS6^−/−^ mice liver [84]. This drop in C16:0 ceramide content was associated with rescued mitochondria morphology. Through proteomic approaches, the authors tried to identify specific sphingolipid–protein interactions that could drive these differences. They reported a specific interaction between CerS6-derived sphingolipids and the mitochondrial fission factor (MFF). Whole-body knockdown of MFF due to shRNA protected HFD-fed mice to the same extent as CerS6 deletion. More importantly, MFF deficiency did not further inhibit mitochondrial elongation in CerS6^iKO^ mice, suggesting that both factors are necessary to promote mitochondrial dysfunction. These data support the findings of Smith et al., who observed enhanced mitochondrial fission in C2C12 myotubes treated with palmitate [98]. Thus, the inhibition of the mitochondrial fission protein dynamin-like protein-1 was sufficient to prevent palmitate-induced mitochondrial fission and IR. These new findings underscore a direct causal role of ceramides in the regulation of mitochondrial integrity and activity. This incriminates ceramides derived from a specific CerS isoform, whose subcellular localization would be responsible for the generation of a mitochondrial pool of sphingolipids. 

Alternatively, changes in membrane microdomain composition and properties could also modulate insulin signaling and action. Normal insulin signaling requires INSR incorporation in lipid rafts [99,100,101]. Park et al., observed that CerS2 null mouse liver, which has a higher C16-ceramide content, displays a decreased INSR translocation in the detergent-resistant membrane (DRM) fractions upon insulin treatment [88]. Moreover, INSR present in the DRM still displayed impaired insulin-stimulated phosphorylation. It is likely that similar membrane microdomain alterations could also be seen in the mitochondria and account for impaired morphology and activity. Although this research axis has not been extensively investigated, it offers another mechanism by which ceramides inhibit insulin action.

## 4. Conclusions

Collectively, there is a strong body of evidence to support a causal role of DAGs and ceramides in lipid-induced IR. In recent years, many studies using pharmacological inhibitors and transgenic mouse models have, in our opinion, supported the growing interest for ceramides in the pathogenesis of T2D. They incriminated short-chain ceramides (C16–C18) in all metabolic tissues. They evidenced that ceramides could interfere with insulin sensitivity, not only through direct targeting of insulin signal transduction, but also by modulating mitochondrial activity and membrane biophysical properties. Even though ceramides are already known to influence these aspects, and especially mitochondria dynamics, it seems that the accumulation of lipotoxic lipid species has always been considered as a consequence of impaired mitochondrial respiration in the context of IR. However, recent data suggest that ceramides could drive these defects through specific interactions with regulatory proteins. Thus, perturbations in cellular sphingolipid metabolism could influence mitochondrial reactive oxygen species and acylcarnitine production, thus providing a unifying theory for lipid-induced IR. The variety of mechanisms by which ceramides might impair insulin signaling led some authors to propose an attractive evolutionary theory by which ceramides accumulation would signal free fatty acid (FFA) oversupply and promote the preferential incorporation/neutralization of newly imported FFAs into TAG pools to protect the cell from their detergent-like properties [81]. Future understanding of the mechanisms linking ceramides to IR requires the development of new mouse models targeting both ceramides synthesis and recycling pathways, investigating sphingolipid subcellular localization and flux, and identifying new interacting proteins that could be responsible for the induction of specific gene programs, especially related to mitochondrial function. Transgenic mouse models also highlighted crosstalk between peripheral metabolic tissues, particularly the liver and AT. Decreasing ceramide content in AT triggered beneficial effects in the liver. However, this crosstalk remains poorly investigated and future studies will bring new insights by determining how tissue-specific depletion/accumulation of ceramides reciprocally influences remote organs and whole-body phenotypes. This critical analysis also reveals that subcellular pools of specific DAG and ceramides species should be considered when analyzing insulin-resistant phenotypes in other contexts than obesity such as aging and physical inactivity [87,91].

## Figures and Tables

**Figure 1 ijms-21-06358-f001:**
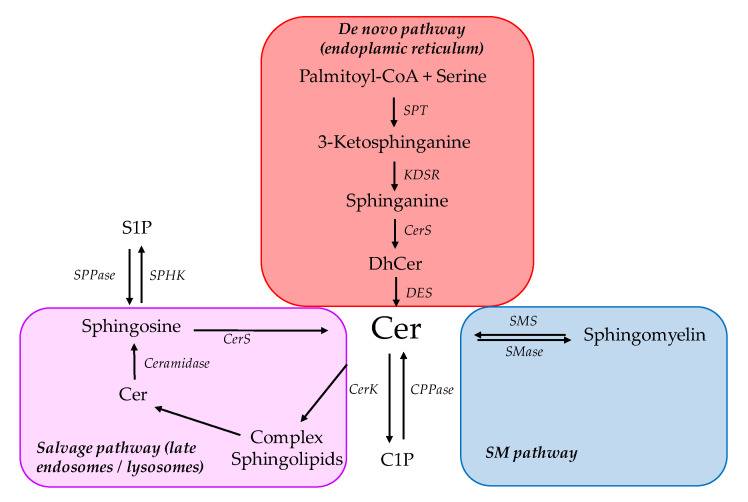
Ceramide biosynthesis pathways. Simplified overview of the main ceramide synthesis and degradation pathways. C1P Ceramide-1-phosphate; Cer Ceramide; CerK Ceramide kinase; CerS Ceramide synthase; CPPase Ceramide-1-phosphate phosphatase; DES Dihydroceramide desaturase; DhCer Dihydroceramide; KDSR Ketosphinganine reductase; S1P Sphingosine-1-phosphate; SM Sphingomyelin; SMase Sphingomyleinase; SMS Sphingomyelin synthase; SPHK Sphingosine kinase; SPPase Sphingosine-1-phosphate phosphatase; SPT Serine pamitoyltransferase.

**Figure 2 ijms-21-06358-f002:**
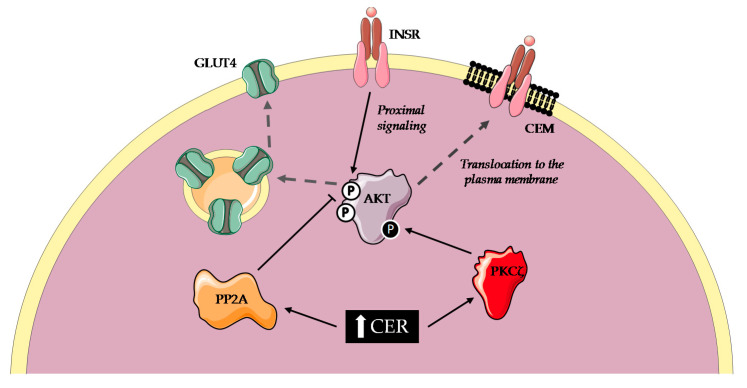
Molecular mechanisms of ceramides-mediated insulin resistance. Short term accumulation of ceramides leads to insulin resistance in various cell types. Ceramides activate PKCζ that phosphorylates Akt and prevents translocation to the plasma membrane. They also activate PP2A which dephosphorylates Akt, in turn blunting Akt-mediated activation of GLUT4 translocation to the plasma membrane. AKT Protein kinase B; CEM Caveolin enriched micordomains; CER Ceramides; GLUT4 Glucose transporter type-4; INSR Insulin receptor; PKCζ Protein kinase Cζ; PP2A Protein phosphatase 2A.

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
