# Peer review of "Novel Insights and Mechanisms of Lipotoxicity-Driven Insulin Resistance"

_ijms, 2020, doi:10.3390/ijms21176358_

Round 1

Reviewer 1 Report

The review is solid and well written but it lacks three important aspects:

1) The Authors in the final part of the manuscript write: "In summary, new data emerging from thorough in vivo studies have built a strong body of evidence for a causal role of ceramides in lipid-induced IR. The variety of mechanisms by which ceramides might  impair insulin signaling led some authors to propose an attractive evolutionary theory by which ceramides accumulation would signal free fatty acid (FFA) oversupply and promotes the preferential incorporation/neutralization of newly imported FFAs into TAG pools to protect the cell from their detergent-like properties [79]". But the weakness of this review regards the role of FFA not even mentioned in the rest of the paper and there is a long but unclear relation between FFA and insulin resistance that the Author must address in a subchapter to better understand this whole fatty-acid-related picture.

2) Insulin is anti-lipolytic and increase TAG content by increasing NPRC expression and thus inhibiting natriuretic peptide induced lipolysis. Does IR reduce this anti-lipolytic effect of insulin?

3) This review badly needs one or more figures summarizing the main mechanisms described in the text 

Author Response

The review is solid and well written but it lacks three important aspects:

1) The Authors in the final part of the manuscript write: "In summary, new data emerging from thorough in vivo studies have built a strong body of evidence for a causal role of ceramides in lipid-induced IR. The variety of mechanisms by which ceramides might impair insulin signaling led some authors to propose an attractive evolutionary theory by which ceramides accumulation would signal free fatty acid (FFA) oversupply and promotes the preferential incorporation/neutralization of newly imported FFAs into TAG pools to protect the cell from their detergent-like properties [79]". But the weakness of this review regards the role of FFA not even mentioned in the rest of the paper and there is a long but unclear relation between FFA and insulin resistance that the Author must address in a subchapter to better understand this whole fatty-acid-related picture.

We thank the reviewer for raising this important point. We have now included a few additional sentences in the introduction section to clarify these aspects and a novel reference (PMID: 18177724).

In fact, there is little evidence that FA per se trigger defects in insulin signaling and induce insulin resistance. Excess FA delivery and uptake in peripheral tissues such as liver and skeletal muscle will lead to excess intracellular FA flux. These FA will first be activated to form fatty acyl-CoA and then further converted into triacylglycerols (TAG) within lipid droplets [10,11] or give rise to diacylglycerols (DAG) and ceramides through specific metabolic pathways that will be detailed in the following chapters of this review. FA overload to the mitochondria can cause also cause incomplete oxidation of fatty acyl-CoA thus producing an excess of acyl-carnitine that has been associated with defects in insulin signaling, although the precise molecular mechanisms involved still remain unclear (PMID: 18177724). In addition, as reviewed in details elsewhere, an inability to adequately store the excess FA influx into TAG-lipid droplets precipitate lipotoxicity [10].

2) Insulin is anti-lipolytic and increase TAG content by increasing NPRC expression and thus inhibiting natriuretic peptide induced lipolysis. Does IR reduce this anti-lipolytic effect of insulin?

We did not discuss this point in the manuscript. Although interesting we believe this point is out of scope of the purpose of the review. As a matter of discussion, we attempted to address this point hereafter.

Insulin is indeed the main lipolytic hormone in humans. A study by Pivovarova et al. (J Clin Endo Metab 2012; 97:731-739) reported that insulin up-regulates natriuretic peptide clearance receptor (NPRC) expression in human adipose tissue. We agree this could contribute to blunt natriuretic peptide-mediated lipolysis in the fed state. There is still a debate on the role of insulin resistance in adipose tissue. Some investigators see this as a biological adaptation to avoid excess fat storage. Others see this as a contributing factor to whole-body insulin resistance by releasing a break on the anti-lipolytic effect of insulin and favouring excess fatty acid spill over and possibly local inflammatory events within adipose tissue.

3) This review badly needs one or more figures summarizing the main mechanisms described in the text 

Thanks for this suggestion. We have now included two novel Figures in the revised manuscript as requested.

Reviewer 2 Report

The purpose of the review is to summarize and discuss the most recent studies supporting a causal role of the total pool or specific levels of lipotoxic lipids such as diacylglycerols (DAG), ceramides and sphingolipids in the pathophysiology of insulin resistance (IR) with special attention given to the transgenic mouse models available which regulate tissue-specific intracellular levels of these complex compounds.  

We commend the authors for attempting to summarize a vast topic for the readers of the Journal. In view of the complexity of this topic and the amount of information the authors attempt to summarize, we would suggest that they format this article in a  more user-friendly fashion such as what was done in a recent review of the topic by Sokolosaka et al. 1

We have the following comments and suggestions.

In line 15 the term insulin resistance is introduced without the (IR) abbreviation. (IR)

In line 17, 19, 23, 35, IR should be used instead of insulin resistance

In line 54 please use the term “guilty as charged” in quotations

In line 58 please correct the spelling of signalling

In line 67 there is no need to capitalize Insulin

In line 68 please correct the spelling of signalling

In line 71 please correct the spelling of signalling

In line 74 please correct the spelling of signalling

In line 99 please complete the sentence “Interestingly, authors also reported a positive relationship between mitochondrial/endoplasmic reticulum (ER).”

In line 113 the sentence needs a reference.

In line 105 an explanation or reference for the Kennedy pathway is needed.

In section 3. Causal role of ceramides in insulin resistance? A figure depicting the complexity of ceramide and sphingolipid pathways discussed that section may be necessary for the readers. 

In line 134 there is a typo liming should be limiting step line 134

In line 134 Serine does not need to be capitalized.

In line 141 the abbreviation CERT is introduced without its corresponding Ceramide transfer protein CERT .

In line 157 the abbreviation PP2A is also introduced without its corresponding Protein phosphatase 2A

In line 159 similar abbreaiation CEM is introduced without its corresponding Caveolin-enriched microdomains (CEM)

In line 169 the Degs1 deficiency on insulin sensitivity should identify the two studies with references.

In line 182 there is a typo that should be corrected. Involve should be replaced by involved.

The reference 77 is repeated as it is the same as 76.

In line 191 the signaling cascade defects that are observed in IR could use a figure or a summary.

Line 238 states the mice were sacrificed after 12 weeks of HFD and  239 myriocin treatment, therefore limiting body weight gain [77].     

In line 243 the sentence “Overall, the most interesting finding was that myriocin enhanced a thermogenic program in sWAT, 243 but not eWAT, where ceramide levels were not reduced. Needs a reference>

In line 259 is the author using the correct reference?

In line 269 the authors state two studies investigated the effect of Degs1 deficiency on insulin sensitivity but they do not list the two references. Are they referring to reference 71 and 79?

In line 466 the authors should list the references they are referring to.

In line 480 the authors refer to which reference? They introduce the term MAM and do not give it a definition. Mitochondrial associated membrane

Section 3.4 Emerging mechanisms of action of ceramides introduces a number of mitochondrial and membrane specific mechanism which need to be outlined in a more consistent manner.

Line 503 is a summary statement that belongs in section 4 Conclusion

  1. Sokolowska E, Blachnio-Zabielska A. The Role of Ceramides in Insulin Resistance. Frontiers in Endocrinology. 2019;10(577).

Author Response

Reviewer 2:

The purpose of the review is to summarize and discuss the most recent studies supporting a causal role of the total pool or specific levels of lipotoxic lipids such as diacylglycerols (DAG), ceramides and sphingolipids in the pathophysiology of insulin resistance (IR) with special attention given to the transgenic mouse models available which regulate tissue-specific intracellular levels of these complex compounds.  

We commend the authors for attempting to summarize a vast topic for the readers of the Journal. In view of the complexity of this topic and the amount of information the authors attempt to summarize, we would suggest that they format this article in a more user-friendly fashion such as what was done in a recent review of the topic by Sokolosaka et al. 1

Sokolowska E, Blachnio-Zabielska A. The Role of Ceramides in Insulin Resistance. Frontiers in Endocrinology. 2019;10(577).

We thank the reviewer for this suggestion and apologize for omitting this important reference in the first version of the manuscript. We have now included this reference in the revised manuscript (line 57) and formatted in a more friendly fashion by including two figures (schematic models), one on the ceramide synthesis pathways and one on the molecular mechanisms of ceramide-induced insulin resistance. We carefully attempted to avoid overlapping too much with previous reviews on this topic.

The following editing revisions requested have been performed:

  • Line 15: (IR) abbreviation was added
  • Line 17, 20, 24, 35: insulin resistance was replaced by IR
  • Line 54: the term “guilty as charged” was put in quotations
  • Line 58, 68, 71, 74: “signaling” was corrected to “signaling”
  • Line 67: capital I of Insulin was replaced
  • Line 99: the sentence was completed by removing the full stop
  • Line 113: The sentence was reformulated and the appropriate reference was added
  • Line 114: The appropriate reference was added
  • Line 105: The appropriate reference was added
  • Line 134: Typo corrected
  • Line 134: Serine corrected to serine
  • Line 141, 157, 159: the appropriate abbreviations are now cited first time in the text
  • Line 182: Involve replaced by involved
  • Reference 77 and 76 were merged
  • Line 191: A figure has now been included in the revised manuscript
  • Line 238: Sentence reformulated
  • Line 243: Reference added
  • Line 259: The reference given is the appropriate one
  • Line 269: The two references were added
  • Line 466: The sentence is an introduction to the rest of the paragraph and the references are presented sequentially in the rest of the text.
  • Line 480: The appropriate reference was added
  • Section 3.4: Some clarifications have been included.
  • Line 503: The concluding remarks of the paragraph were transferred in the Conclusion